# Intelligent Video Highlights Generation with Front-Camera Emotion Sensing

**DOI:** 10.3390/s21041035

**Published:** 2021-02-03

**Authors:** Hugo Meyer, Peter Wei, Xiaofan Jiang

**Affiliations:** Department of Electrical Engineering, Columbia University, New York, NY 10027, USA; ham2162@columbia.edu (H.M.); pw2428@columbia.edu (P.W.)

**Keywords:** mobile computing, emotion recognition, image processing, signal processing algorithms

## Abstract

In this paper, we present HOMER, a cloud-based system for video highlight generation which enables the automated, relevant, and flexible segmentation of videos. Our system outperforms state-of-the-art solutions by fusing internal video content-based features with the user’s emotion data. While current research mainly focuses on creating video summaries without the use of affective data, our solution achieves the subjective task of detecting highlights by leveraging human emotions. In two separate experiments, including videos filmed with a dual camera setup, and home videos randomly picked from Microsoft’s Video Titles in the Wild (VTW) dataset, HOMER demonstrates an improvement of up to 38% in $F_{1}$-score from baseline, while not requiring any external hardware. We demonstrated both the portability and scalability of HOMER through the implementation of two smartphone applications.

## 1. Introduction

Several years ago, the limits of digital storage compelled people to only keep their best photos and video highlights. Today, the improvement of digital storage capacity, the emergence of the cloud, the advancements in compression standards, and the prevalence of social networks have allowed users to increase their generation and consumption of pictures and videos. According to Cisco, videos are responsible for 75% of global IP traffic, and this proportion is anticipated to increase to 82% in 2022 [1]. The large amount of stored images and videos makes compiling a digital photo album or creating a montage a greater challenge. Creating a montage represents a significant time investment, requiring selection of segments to keep only the highlight moments. Furthermore, abundant storage capacities encourage the recording of longer videos, thus increasing the proportion of non-highlight segments in a video. Finally, home editing software such as Adobe Premiere, Apple iMovie, or CyberLink are only accessible to users with prior editing knowledge and computer skills [2], considerably narrowing down the targeted market.

The focus of this work is to automate the generation of video highlights, thus saving time for people manually developing their own montages and scanning through hundreds of media files to find video highlights. Besides home users, automatic video highlight generation can also lead to a wide range of other applications such as interactive browsing [3,4], searching systems [4], and sports match summaries [5,6,7,8].

Facebook, Apple and Google have developed similar technology, with limited success. Facebook’s “One year in review”, a feature summarizing the user’s year based only on uploaded photos and videos, considerably constrains the available content. This automatic and unsupervised generative feature also raised ethical questions due to the Al-Qaeda scandal [9], where “One year in review” produced terrorism propaganda. Both Google’s “Creations” and Apple’s “For you” features attempt to piece together videos and photos to represent the happiest memories, but often uses content with low relevance. These feature algorithms appear erratic, often drowning short video segments amongst photos, fancy transitions, and background music. No interface exists for the user himself to enter any input.

In the field of video summarization, external webcams are utilized to detect user’s facial expressions [2,10,11], eye blinks, eye movements, and head motion [2]. These works require the use of fixed cameras and suffer from immobility, significantly restricting the range of potential applications. The requirement of using obtrusive and non-wearable sensors to extract a user’s emotions has encouraged most research to focus on direct affective content utilizing video frames [12,13,14,15], video segments [16,17,18,19], audio and text features [14]. The present results remained poor on average due to the semantic gap problem [10] of detected objects or events.

As opposed to video summarization, where the aim is to summarize an entire video, this work focuses on extracting video highlights based on affective content, which might not be present in emotionally neutral videos where no stimulating event occurs. Wang et al. [20] describe affective content as either direct content, corresponding to data extracted from the original video stream, or implicit content, corresponding to the user’s spontaneous response when watching the video. As video highlights tend to be subjective, implicit content is a relevant feature for automatic extraction. Previous research has leveraged different physiological signals to detect a user’s emotions for video emotion tagging, such as EEG signals [21,22,23], electrodermal activity [24], pupil response [22], and head motion [25]. The limitation of these approaches is the use of external hardware such as obtrusive and non-mobile sensors. This work’s approach does not require any additional hardware other than a smartphone.

This work addresses the lack of high-quality video highlights generation with HOMER (illustrated in Figure 1), a system that automatically extracts video highlights using a combination of emotional features of the user and audio-visual features in the video. The user is able to specify personal preferences through a web API and customizable parameters. The algorithms that we developed outperform state-of-the-art solutions by utilizing a user’s facial expressions, video frames, and video audio.

Our paper presents the following key contributions:A novel multi-modal system combining human emotion and audio-visual features to achieve an intelligent, personalized, and user-oriented highlight extraction.The design of a scalable event timeline to coherently fuse extracted features and enable the synchronisation of heterogeneous input streams.A novel algorithm mapping event timeline features into two inter-dependant score functions for optimal highlight cuts.The usage of video frame similarity to detect dynamic variations in a video for enhancing the generalization of highlight content.A high detection accuracy of sound events by utilizing the wavelet transform and a bottom-up peaks clustering algorithm.

## 2. Background

Money and Agius [26] divided video summarization techniques (which applies analogously to video highlight extraction) into three categories: internal techniques utilizing data directly sourced from the video stream, external techniques utilizing data not directly sourced from the video stream, and hybrid techniques (both internal and external).

### 2.1. Internal Content Techniques

Internal content video summarization is the broadest and most visited category because of its high scalability and easy implementation. No sensor or any other source of information is required; only the video itself is used as input. Internal content highlight detection was performed extensively in the context of sports because of the high user demand, the diversity of video features (slow-motions, shots pace, ball and players tracking, etc.), and the relevance of audio (energy strongly correlated to excitement, extraction of keywords, etc.) [5,6,7,27,28].

Early research extracted audio-visual features and used self-designed algorithms to transform them into segment boundaries, often derived from a resulting score function [26]. The self-designed algorithms progressively led to classifiers [20] and a recent emergence of full deep learning implementations. Supervised deep learning techniques use human-set annotations, which correspond to a limited resource and are often subjective, making the task non-trivial [13,15,16,29,30,31,32,33,34,34]. Unsupervised techniques do not require any complementary annotations. They can be trained on larger amounts of data, which often improve the reliability and the generality of the model [17,18,35,36,37,38,39,40]. Although these works have shown good results in video summarization and highlight generation, these works rely on features from audio and video. Due to this restriction, personal highlight generation is not possible.

### 2.2. External Content Techniques

Although most of the works such as [41] use external affective content for focused applications such as emotion tagging in video, Joho et al. [10,11] utilized the viewer’s facial expressions for video summarization. Despite the low performance of their model, they are able to use an unobtrusive sensor to retrieve affective content. On the other hand, Chênes et al. [42] proposed a solution based on different physiological signals such as electromyogram (EMG), blood volume pulse (BVP), electrodermal activity (EDA), and skin temperature. Their positive results are overshadowed by the necessity of obtrusive senors, which remains a significant limitation. Overall, very few applications using external affective content exist in the field of video summarisation and highlight extraction.

### 2.3. Hybrid Content Techniques

Hybrid content techniques must deal with the constraints of external content sensing and the fusion of external and internal information. A hybrid affective approach was driven by Peng et al. [2], who proposed to model both the user’s attention and the user’s emotions in addition to video shot detection. The user’s attention was inferred by detecting head motion, eye blinks, and eye saccades, while the user’s emotions were derived from facial expressions. As in [10], video segments are extracted directly by maximizing a user emotional score function. This is a limited approach in cases where emotional response to a stimulus shifts in time. Their score is only based on user satisfaction, making the comparison difficult. Fiao et al. [43] also uses emotions along with video to generate highlights; however, they focus on sports videos, and requires external hardware and human interaction to gather emotion data. Ringer et al. [44] used deep learning to measure emotion of a face camera during streaming of video games; this work is restricted to highlight the generation of video games, which similarly to sports, is strongly correlated with audio energy. Kaklauskas et al. [45,46] study specifically video ads, and require a plethora of hardware and human interaction to gather various physiological signals.

State-of-the-art solutions are a significant improvement in the field of internal video summarisation, largely due to deep learning breakthroughs. However, the absence of user emotion perception makes the existing models well-adapted for video summarization, but irrelevant for the extraction of highlights. The high subjectivity of the highlight semantic makes external content-based techniques more suitable for the task. The ideal case of hybrid solutions is rarely visited, pushing us to develop a hybrid solution.

In summary, we identified four types of limitations present in current literature in automated highlight generation. The first limitation is the scope of videos analyzed; focus on a specific type of video, such as sports or video games, limits the generalizability of these works to a specific setting, and may not work well for videos outside of that setting. The second limitation is a reliance on audio or visual features; relying solely on the video content can reduce the personalization of highlights which can be different between people. The third limitation is the usage of external hardware, which can be cumbersome to setup or requires human interaction with the system. Finally, this is the first system developed on a mobile platform, which is not only a popular medium for capturing video, but also has the potential for more widespread dual camera video capture in future generations of mobile technology. The limitations of relevant existing works in highlight generation are shown in Table 1 and Table 2.

## 3. System Design

In this section, an overview of the overall system architecture is presented, followed by a detailed stage-wise analysis of each component.

### 3.1. System Architecture

Due to a combination of computational requirements for video processing and our desire to create a web service that can enable multiple third party applications, HOMER is currently implemented on a cloud server rather than a mobile device. As there is no real-time requirement for our applications, a cloud server allows for significant energy savings, as well as flexibility in development for future improvements. A major concern of cloud-based solutions is data privacy, especially as sensitive data are sent to the server. We address this concern in HOMER by storing data securely during highlight extraction, and deleting all data immediately after. In future works, pre-processing (such as obfuscation or feature extraction) of sensitive data can be performed on the mobile device to eliminate the need to send sensitive data to the cloud.

As shown in Figure 2, the web-service API allows the client to provide an input stream composed of input signals and parameters. The two required input signals are the original video (scene video) and a synchronized video (facial video) of the user’s face. The facial video is an important input, allowing for emotion recognition without requiring external hardware. Firstly, the scene and facial video frames and scene video audio are extracted and used as separate raw signals. Secondly, the three raw signals are pre-processed and subjected to feature extraction. The features extracted are then fused and synchronized using our events timeline (ET) concept. Next, an algorithm was specifically designed to convert the ET features into two score functions, from which both highlight start and end times are derived. Finally, the generated highlight (output stream) is sent back to the client via the web service API.

### 3.2. Input Stream

The input stream provided by the client is composed of the following signals and parameters:*Scene Video*: video to be highlight-extracted. Both video frames and audio signals are extracted as distinct raw signal inputs for the highlight extraction pipeline.*Facial Video*: video of the user’s face recorded in reaction to the recorded scene. Its duration must be equal to the scene video duration.$T_{HL}^{min}$: parameter indicating the desired highlight minimal duration. If set to −1, the algorithm will automatically decide the highlight length.$T_{HL}^{max}$: parameter indicating the desired highlight maximal duration. If set to −1, the algorithm will automatically decide the highlight length.$\gamma_{mult}$: boolean indicating whether multiple highlights should be generated.

### 3.3. Signal Pre-Processing and Feature Extraction

The three input signals (facial video frames, scene video frames, scene video audio) should be pre-processed before feature extraction to generate homogeneous data that can be fused to make highlight decisions.

#### 3.3.1. Emotion

As shown in previous works [23,42], a viewer’s emotions while watching a video help indicate the location of emotional highlights. This was also demonstrated by Joho [10], who achieved video highlight segmentation based exclusively on viewer’s facial expressions. In this work, we chose to utilize users’ facial expressions as a key feature and focused on positive emotions only under the assumption that users are more willing to keep positive moments of their lives. In this way, we can capture personalized highlights while capturing important features from the scene video, which may not be indicated by emotional responses alone as in [10]. To extract emotion probabilities from frames of the user’s face, two computer vision steps were performed:

(a) Face Detection.

In some cases, the user’s full face may not be visible in the frame. HOMER utilizes face detection to improve the emotion recognition model through data standardization. Using the Haar-cascade open-source face detection algorithm, the user’s face is cropped from the raw image in each frame (Figure 3a). In cases where no face is detected, the unknown emotion probabilities are linearly interpolated.

(b) Emotion Recognition. Most state-of-the-art models achieve emotion recognition through two different approaches: classifying extracted action units (AUs) of the human face [50,51] or performing both feature extraction and classification using deep learning [52]. After comparing different available trained and open-source models, we opted for piyush2896’s deep learning solution (https://github.com/piyush2896/Facial-Expression-Recognition-Challenge) to the Facial Expression Recognition Challenge on Kaggle. The model achieves a desirable compromise between computational time and performance, with an accuracy of 69% over seven classes (fear, anger, sadness, disgust, neutral, happiness, surprise) on the AffectNet dataset [53]. As shown in Figure 3b, we only detect positive emotions and simplified the problem to only three classes: happiness, surprise and all other emotions ($\equiv 1-p_{hap}-p_{surp}$).

(c) Peak Detection. From the two resulting signals (happiness and surprise), specific event features had to be extracted. A peak detection algorithm was developed to extract peak sides and peak top (Figure 3c). As the model presented varying sensitivities to emotions, different adapted thresholds were used for happiness and surprise.

(d) Feature Extraction. Each emotion peak represents an event and is described by five features: the start of peak rise, the start of peak top, the end of peak top, the end of peak fall and the average value of peak top. The resulting simplified signal from this feature extraction is shown in Figure 3d.

#### 3.3.2. Video

HOMER utilizes the frame similarity (FS) between consecutive frames of video to improve highlight detection robustness. Frame similarity was used in order to detect dynamic state changes throughout the recording, such as differentiating static camera and camera pan. This feature improves the precision of the highlight boundary decisions by enforcing the boundaries to correspond to the start and end of a camera dynamic mode. For instance, if a camera pan occurs at some point in the video, the start and end of the pan will be prioritized over other timestamps. Frame similarity also helps to increase highlight content diversity by prioritizing instances where the camera moves between two scenes, resulting in a highlight based on the first scene, content between both scenes, and the second scene. An example of the corresponding frame similarity pattern is shown in Figure 4e between the two purple dots.

To detect these dynamic changes as one-time point features, the following steps were performed:

(a) Frame Extraction. The frames are extracted from the video with a higher rate than emotion recognition due to the fast possible variations of the signal (Figure 4a).

(b) Frame Similarity Computation. Frame similarity was computed pixel-wise using the work done by Zhang et al. [54]. This method focuses on pixel value changes every three consecutive frames in the binned HSV domain (Figure 4b).

(c) Discontinuity Removal. As shown in Figure 4b, the computed signal may present short and significant drops, which are barely visible in the original video. These artifacts may be due to luminosity variations, which make the pixel values change enough to be detected as proper changes. These discontinuities often present similar properties, which facilitates removal with simple shape conditions (Figure 4c).

(d) Low-pass Filter. The signal is smoothed using a low-pass filter with a cutoff frequency of 2.5Hz. The resulting signal provides an alternated succession of local maxima and minima.

(e) Feature Extraction. We define the pre-processed frame similarity function as:S(k),k∈K
where *K* is the total number of frames. We identified all the local maxima of *S* as follows:$$Q=\{k\;|\;S^{\prime }\left(k\right)=0,\;S^{\prime }\left(k-1\right)>0,\;S^{\prime }\left(k+1\right)<0\}$$

To build the final set *F* of feature points, only the local maxima subject to high variations were kept with the following condition:$$F=\{q_{i}\in Q\;|\;\max\left(S\left(q\right)-S\left(x_{L}\right),\;S\left(q\right)-S\left(x_{R}\right)\right)>\sigma \}$$
where $x_{L}$ is the frame index of the local minima on the left of $q_{i}$, $x_{R}$ is the frame index of the local minima on the right of $q_{i}$, and σ is a threshold value.

Each feature point is associated with a score proportional to both the total amount of local variation and the difference between right and left variations. Finally, we observed that the main patterns of similarity have varying importance. The score is re-scaled accordingly to these particular categories using fuzzy logic.

The three commonly encountered patterns are (Figure 5):“Hills”: these frame similarity peaks translate either to brief camera stop-motions or noise, and thus are assigned low importance.“Plateau”: this pattern translates to an extended camera stop, often due to an event of interest. This frame similarity shape also appears when a camera zoom-in/zoom-out is performed. The medium level of importance was chosen instead of high because of the lack of summarisation this pattern offers. Only one scene of the video would be contained in the generated highlight, versus two for the “Valley” pattern (see below).“Valley”: this pattern often translates to a camera pan or tilt, from one motion stop to another. This pattern was assigned high importance for its semantic generalization. This pattern contains information from the first motion stop scene, information from the second motion stop scene and information between both scenes. Furthermore, the in-between content often presents important information, due to the constant low similarity value. As an example, this pattern occurs when the user is performing a constant pan/tilt to show an area or a beautiful landscape.

Each frame similarity feature point is now fully described by two parameters: its frame number $f_{i}$ and its adjusted score $C_{f_{i}}^{\prime }$. An example is shown in Figure 4e.

#### 3.3.3. Audio

The sound of a video is a meaningful indicator of events and can deliver additional information about the video semantics and affective content. Many state-of-the-art techniques combine event detection and event recognition into deep learning models, sorting out polyphonic sound event classes [55,56,57]. These models are usually complex, computationally heavy and often intended to solve high multi-class problems, that lead to poor results and non-trivial processing for our highlight extraction model. After watching numerous user videos, we note a strong correlation between the relative energy of sound events and the emotional level. In addition to sound energy, the nature of the sound itself is meaningful information. For example, a car door slamming should be considered as less important than a scene where an actor suddenly starts laughing. All these considerations lead us to perform both monophonic sound event detection and low multi-class sound event recognition, which are later combined into final feature events.

(a) Event Detection: The sound signal of a video can be segmented into different events based on amplitude. The higher the sound energy is, the more likely that the content will be important or affective. To detect and segment high-energy events relative to the rest of the signal, we pre-processed the signal accordingly and designed a bottom-up unsupervised event extraction (Figure 6).

Pre-processing: Firstly, sound samples which only contain high amplitude noise are discarded. To detect such cases, the energy density of the whole sound sample is computed. This metric is independent from the sample size:$$W=\frac{1}{T}\int_{0}^{T}{\left|s\left(t\right)\right|}^{2}\;dt,\;0\leq t\leq T$$

Any sound sample having *W* above the threshold σ of 0.7 is considered as containing no event and is discarded. Squaring the signal generates positive values and allows differentiation between low-interest and high-interest events.

After normalization, we use the wavelet transform to transform the signal u(t) into a lower frequency signal $u_{0}\left(t\right)$, with a similar shape to the envelope of u(t). This new signal $u_{0}\left(t\right)$ is better suited for handling the detection of events. The one-dimensional fast wavelet transform (FWT) was decomposed using 7 levels (J = 7) and derived using the Daubechies-4 mother wavelet. After a second normalization, the signal is adjusted by its median, removing any remaining offset caused by background sounds. The result of the pre-processed sound signal is shown in Figure 7b.

*Event extraction:* Sound event extraction is performed using an unsupervised bottom-up approach. After pre-processing, the sound wave is transformed into a succession of peak-shaped sub-events. To extract events, relevant sub-events are clustered into meaningful high-level events using a threshold Θ on the pre-processed signal. We use a flexible and adaptive approach to determine Θ, as shown in Algorithm 1. The number of crosses per second is first computed between a sweep of Θ (from 0 to 0.5) and the pre-processed sound signal (Figure 7b), resulting in Figure 8. As a first step, we ensure that the signal is non-null. A second check determines whether an elbow shape (defined with specific criteria) appears when increasing the value of Θ (teal circle). The presence of an elbow represents a large group of sound sub-events sharing a similar amplitude. The sub-events likely to be background sounds are discarded by setting Θ at the elbow. In the absence of an elbow, if the maximal number of crosses remains under a certain threshold value (red dashed line), Θ is set to the lowest possible value above zero, otherwise we increase Θ until the number of crosses is below the threshold. Sub-events are determined by considering the signal above threshold Θ. The boundaries of each sub-event are determined when the signal crosses the Theta, as in Figure 7b.
**Algorithm 1** Adaptive threshold computation**Require:** sound signal of n values**Ensure:** adaptive threshold value Θ1:$C_{arr}$← Nb of crosses between S and Θ, Θ ranging from 0.01 to 0.52:$C_{max}=\max\left(C_{arr}\right)$3:$\Theta_{max}=argmax_{\Theta }\left(C_{arr}\right)$4:**if**$C_{max}==0$**then**5: **return** 06:**end if**7:**if** elbow present in $C_{arr}$
**then**8: **return**
Θ at the elbow9:**end if**10:**if**$\max(C_{arr})>2$**then**11: **return**
min(Θ) s.t. $C_{arr}<2,\;\Theta_{max}<\Theta$12:**else**13: **return** lowest Θ (0.01)14:**end if**

The sub-events are subsequently clustered into high level events by Algorithm 2. The algorithm takes three primary inputs: the maximal time gap allowed one to merge two sub-events ($\Delta t_{th}$), the maximal score difference allowed to merge two sub-events ($\Delta S_{th}$), and the maximal quiet event duration allowed to be merged with a louder event ($T_{max}$).

The algorithm is divided into two loops:

The first loop (lines 3–10) progressively merges sub-events by comparing sub-event $d_{i}$ with the current growing cluster of sub-events (Figure 7b). Once a stop condition is reached (line 4), the current cluster is closed and a new cluster is opened with sub-event $d_{i}$.

The second loop (lines 11–16) considers every two consecutive cluster events. The two clusters merge into a bigger event if they meet the time distance condition and either meet the score distance condition or if the event with lower score has a duration below the threshold $T_{max}$. Finally, a safety check ensures that the duration of all extracted sound events does not exceed a percentage of the total video duration. As sound events are features used in the inference of a highlight, important events may be overshadowed by less important ones if the percentage is too high. The reliability of the sound feature for highlight extraction is then diminished. In the case of high coverage (>0.7), the process returns to the search of sub-events (Figure 6) by increasing the adaptive threshold Θ. Each final event feature is fully described by three parameters: event start, end and score (corresponding to the event energy density).
**Algorithm 2** Sound events clustering**Require:** sub_events D, $\Delta t_{th}$, $\Delta y_{th}$, $T_{max}$**Ensure:** events E1:y = amplitude of a sub-event/event2:X ← first cluster to be grown, initialized with $d_{0}$3:**for** each sub_event $d_{i}$ of D **do**4: **if** ($t_{X}-t_{d_{i}}$ < $\Delta t_{th}$) & ($|y_{X}-y_{d_{i}}|$ < $\Delta y_{th}$) **then**5:  X ← X + $d_{1}$6:  y ← new weighted amplitude of X7: **else**
8:  insert $d_{i}$ into E as a new cluster to be grown9: **end if**
10:**end for**11:**for** each pair ($E_{i}$, $E_{i+1}$) in E **do**12: $E_{low}$← event of ($E_{i}$, $E_{i+1}$) with lowest score13: **if** ($t_{E_{i}}-t_{E_{i+1}}$ < $\Delta t_{th}$) & ($|y_{E_{i}}-y_{E_{i+1}}|$ < $\Delta y_{th}$||$T_{E_{low}}<T_{max}$) **then**14:  Merge $E_{i}$ with $E_{i+1}$, the new y being weighted with respect to events widths15: **end if**
16:**end for**

(b) Event Recognition: To provide more information on the segmented events, a sound event classifier was used to label each event with one of three labels: speech, laughter, or miscellaneous (Figure 9). Speech and laughter were identified as relevant sound events that were often associated with important, highlight-worthy moments.

A speech versus non-speech binary classifier and a laughter versus non-laughter binary classifier were combined to form the three-class classifier in a one-versus-all strategy. Each binary classifier includes two parts, adapted from Wagner et al. [58]: first, a three-layer convolutional network, and second, a two-layer gated recurrent units (GRU) network, where the convolutional layers served the role of feature extraction from the raw audio spectrogram.

This approach was chosen over conventional feature sets such as the mel frequency cepstral coefficients (MFCCs), because the feature representation learned by the convolutional network demonstrated greater robustness with real-world noisy audio data, corroborating work done by Choi et al. [59].

(c) Segmentation and Recognition Fusion: Event detection and event recognition are combined under the assumption that a detected event is homogeneous, containing one class only. Based on majority vote, a segmented event is labelled with the predominant class, as shown in Figure 10. An event is now described by start/end times, score and label (speech, laughter or miscellaneous).

### 3.4. Feature Fusion

In existing research, fusion of multi-modal features is performed on non-time dependant features extracted from time-dependant signals. These approaches do not require synchronization between features, and are typically thrown into classifiers, such as support vector machines (SVM) [60,61], Bayesian networks (BN) [21,62], hidden Markov models (HMM) [28,63], or neural networks (NN) [64]. HOMER performs time dependent feature fusion to retain important information about the timing of certain events.

#### Events Timeline

To provide a precise highlight detection using time dependent features, we extract features in the time domain, requiring synchronization in time. We designed the events timeline (ET) to facilitate the synchronization of the three input signals (emotion, video and audio). As presented in Figure 11, the ET is a matrix of size M×N, where *M* is the total number of features derived from sound, frame similarity and emotion and *N* the number of timestamps (N= fps × duration). Every extracted feature from each of the signals (Section 3.3) is projected onto the ET, as either a class or a score (from 0 to 1). For instance, emotion boundary classes are divided into ”rise start”, “top start”, “top end” and “fall end”. The score represents the importance of an event for highlight extraction. The standardized matrix form of the ET is highly convenient for further computations. The ET facilitates two different approaches to highlight extraction, which will be discussed in Section 3.5.

### 3.5. Highlight Extraction

The event timeline is used to decide the start and end times for highlights. We implemented two different approaches in HOMER: a machine learning approach and a hand-designed approach.

#### 3.5.1. Machine Learning Approach

We present the model design, analyze the results obtained, and discuss the limitations of machine learning in this application.

(a) Model Design: A machine learning solution is a convenient option, as a deep understanding of the causality between features and output is not required. Especially in the case of highlight extraction using multi-modal inputs, constructing a hand-designed model is a non-trivial task. The main challenge is to manually annotate video highlights to supervise the model training. Each video may contain one or more highlights, labelled with start and end timestamps. A sliding window of 5 s is used to build fixed-size inputs from the ET, each containing exactly one ground truth label (either highlight start or end).

In contrast to previous works, where video highlight/summaries are generated with a single score function representing frame-level importance [2,10,12,13,14,15,18,19], we trained two independent classifiers on highlight start and end times. This divide-and-conquer approach simplifies the task for each classifier by specializing in only one classification (Figure 12).

The classification is multi-class (20 classes), each class being a time-stamp within the window input but with a lower fps. Fewer output classes than input points allows for easier learning and decreases the required model capacity. During testing, each classifier outputs a start/end probability score for each class, resulting in two score functions (Figure 12).

(b) Classification Results: Each classifier was trained using 2000 input samples derived from 300 labelled videos. Different models were used initially to determine whether machine learning was an appropriate approach: logistic regression (LR), support vector machine (SVM), k-nearest neighbors (kNN), random forest (RF) and adaboost. The results of the highlight start classifiers are presented in Table 3; the highlight end classification produces similar results. The results were also computed with slack, allowing a classification error of ±0.5 s.

The results are globally poor and perform slightly better than a dummy classifier.

(c) Machine Learning vs. Hand Designed Algorithm: Supervised machine learning models are able to learn complex nonlinear problems, with the only requirement being sufficient training data and their associated ground truth labels. To the human operator, a deep understanding of the input/output causality is not required, as long as the causality exists and the operator is able to derive the output of a given input. Nevertheless, the function learned by the model, which tends to describe the reality as faithfully as possible, is a series of deterministic computations. The ground truth labels of the classifier, start/end of highlights, were manually set by looking at the original scene video. Given that highlight labelling is a subjective task, the labels significantly differed from one subject to another. This low input/output causality led to the poor performance of the machine learning classifiers.

#### 3.5.2. Hand-Designed Algorithm

Similar to the machine learning model, the empirical hand-designed algorithm extracts highlights by computing two start and end score functions respectively called $f_{\alpha }$ and $f_{\beta }$. To take the contribution of each feature into account (Figure 13a), specific functions were designed for each and linearly combined into the final score functions. The optimal highlight in the set of highlight candidates H has its start/end ($\alpha_{opt}$, $\beta_{opt}$) at times ($t_{\alpha_{opt}}$, $t_{\beta_{opt}}$) found as follows:$$\max_{t_{\alpha },t_{\beta }}f_{\alpha }\left(t_{\beta }-t_{\alpha }\right)+f_{\beta }\left(t_{\beta }\right)$$

(a) $f_{\beta }$ computation (see Figure 13b): in the majority of scenarios, emotions are triggered in response to a stimulus event, rather than in anticipation of an event. Hence, a user’s emotions constitute a strong indication of a highlight end and is the basis for the highlight end score function. The end score function (blue curve on Figure 13d) is a linear combination of the feature scores:(1)$$f_{\beta }\left(t\right)=f_{sound}^{\beta }\left(t\right)+f_{FS}^{\beta }\left(t\right)+f_{emo}^{\beta }\left(t\right)$$

We define the set of sound events E={S,L,M}, where S = speech events, L = laughter events, M = miscellaneous events. The sound function $f_{sound}$ is then decomposed as follows:(2)$$f_{sound}^{\beta }\left(t\right)=g_{S}^{\beta }\left(t\right)+g_{L}^{\beta }\left(t\right)+g_{M}^{\beta }\left(t\right)$$
where both $g_{S}^{\beta }$ and $g_{M}^{\beta }$ sum the pattern shown by the blue line of Figure 13b for each S/M sound event; laughter was treated differently. The expression of $g_{L}^{\beta }$ for L events $E_{i}$ with $t_{a}^{i}$/$t_{b}^{i}$ as start/end times:(3)$$\begin{array}{cc} \; & g_{L}^{\beta }\left(t\right)=\sum_{i}^{L}h_{i}\left(t\right),\\ & where\;h_{i}\left(t\right)=\left\{\begin{array}{cc} 2C\frac{t-t_{a}^{i}}{t_{b}^{i}-t_{a}^{i}}-C & t_{a}^{i}\leq t<t_{b}^{i}\\ -\frac{C}{\tau }\left(t-t_{b}^{i}\right)+C & t_{b}^{i}\leq t\leq t_{b}^{i}+\tau \\ 0 & otherwise \end{array}\right. \end{array}$$
where τ is a time constant and $C\propto score(E_{i})$.

Intuitively, the score functions are set to low values while the sound event is occurring and are set to high values after the event’s end. The rationale is that highlights should not be cut during a S or M event (constant <0 value), but a L event can be more easily cut before the end, justifying the linearly increasing the curve of Equation (Equation 3).

The frame similarity function $f_{FS}^{\beta }$ (Equation (Equation 1)) was designed by setting high values in a short time interval around each similarity event. The higher the FS event score is, the more influence it will have in $f_{FS}^{\beta }$.

The function pattern associated with an emotional event is represented in Figure 13b. Just like for sound events, the score remains high for a certain time after the event occurrence. This setting was chosen after having encountered many scenarios in which the highlight was still running after the emotional peak. Happiness and surprise emotions are treated analogously for the whole highlight extraction.

(b) $f_{\alpha }$ Computation: (see Figure 13c): Considering the relationship between α and β, $f_{\alpha }$ is computed by searching backwards from $t_{\beta }$. We want to maximize both $f_{\alpha }\left(\alpha \right)$ and $f_{\beta }\left(\beta \right)$, so $f_{\alpha }$ is computed for all βs fulfilling $f_{\beta }\left(\beta \right)>0$. $f_{\alpha }$ also incorporates a time feature as a penalty to constraint highlight duration and to integrate feature events.

The start function (orange curve in Figure 13d) is computed as follows:(4)$$f_{\alpha }\left(t\right)=f_{sound}^{\alpha }\left(t^{\prime }\right)+f_{FS}^{\alpha }\left(t^{\prime }\right)+f_{emo}^{\alpha }\left(t^{\prime }\right)-\lambda t^{\prime 2}$$
where $t^{\prime }=t_{\beta }-t$ and λ = time penalty.

A quadratic time penalty was chosen to outweigh the first order functions, appearing when integrating a sound or emotional event (Equation (Equation 5)). This penalty avoids extended highlights in the case of long sound/emotional events, and allows autonomous highlight extraction without pre-specifying any highlight length requirement. The definition of $f_{sound}^{\alpha }$ differs from $f_{sound}^{\beta }$ in Equation (Equation 2), by integrating events through time. Considering *L* events, $g_{S}^{\alpha }\left(t\right)$, $g_{L}^{\alpha }\left(t\right)$ and $g_{M}^{\alpha }\left(t\right)$ are computed analogously:(5)$$\begin{array}{cc} \; & \forall t<t_{\beta },\;g_{S,L,M}^{\alpha }\left(t^{\prime }\right)=\sum_{i}^{L}h_{i}\left(t\right),\\ & with\;h_{i}\left(t\right)=\left\{\begin{array}{cc} -C_{1} & t_{a}^{i}\leq t<t_{b}^{i}\\ \int_{t_{a}}^{t_{b}}C_{2}\;dt & t_{b}^{i}\leq t\\ 0 & otherwise \end{array}\right. \end{array}$$
where $C_{1},C_{2}\propto score\left(Event_{i}\right)$.


*N.B.: because of the variable change t’= $t_{\beta }$-t in Equation (Equation 4), increasing t here (from left to right) results later in decreasing t (from right to left), starting from $t_{\beta }$.*


We chose the integration of sound events to promote successive close events in the timeline. Close events usually belong to the same semantic scene and integrating allows the combination of the whole semantic scene. The score function would then linearly grow for each new event, pushing α in front of the whole group. Otherwise, given that a time penalty is applied, the algorithm would preferably always put α before the first event encountered. The computation of $f_{FS}^{\alpha }$ is similar to $f_{FS}^{\beta }$. No integration is done for this time-punctual feature; the score of an FS event increases each time a pattern is formed (Figure 5).

The pattern used for the emotional events is represented by the teal line in Figure 13c. As emotion changes are usually triggered in response to a stimulus, the probability of a stimulus is higher at the start than at the end of an emotional peak. This relationship explains the choice of a linear increasing function during the emotional peak. A time margin is also added before the emotional peak to account for delayed emotional reaction. The integration over an emotional event extends the region of interest before the event occurrence by maintaining a constant positive score from the event start until the beginning of the video. The optimal pair $\left(\alpha_{opt},\beta_{opt}\right)$ determines the highlight by trimming the video between $t_{\alpha }$ and $t_{\beta }$ (Figure 13d).

As explained in Section 3.2, the user can set the minimal and maximal highlight duration $T_{HL}^{min}$ and $T_{HL}^{max}$ as input parameters. In this case, the time penalty λ value will change accordingly to $T_{HL}^{max}$, with $\lambda \propto \frac{1}{T_{HL}^{max}}$ to better account for highlights with duration up to $T_{HL}^{max}$.

Another input parameter explained in Section 3.2, $\sigma_{cover}$ defines whether multiple highlights exist. In this case, the algorithm is run multiple times. Each new generated highlight $H_{i+1}$ is retained only if the overlap with each previous highlights $\left\{H_{0},H_{1},\ldots ,H_{i}\right\}$ does not exceed 25%. Highlights are generated until a stop condition is met:–The score of $H_{i+1}$ is lower than a threshold value–i+1 exceeds a user-specified maximum number of highlights

The output of the highlight extraction algorithm is either $\left(t_{\alpha_{opt}},t_{\beta_{opt}}\right)$ in case of a single highlight or $\{(t_{\alpha_{opt}}^{0},$$t_{\beta_{opt}}^{0}),$…,$(t_{\alpha_{opt}}^{n},$$t_{\beta_{opt}}^{n})\}$ in case of multiple highlights.

### 3.6. Output Stream

The timestamps $\left(t_{\alpha_{opt}}^{i},t_{\beta_{opt}}^{i}\right)$ of the extracted highlight(s) are used to trim the original scene video on the server. The generated .mp4 files are sent back via the API to the client, as shown in Figure 2.

## 4. Applications

HOMER supports a large range of new applications, as demonstrated through the implementation of two novel applications on an Android mobile device.

### 4.1. Application 1: Automated Highlight Generation Using Dual Camera

A primary application of HOMER is the use of both the front and back camera of a mobile device simultaneously (Figure 14). When a user records a video through this application using the back camera, a synchronised video of the user’s face is recorded using the front camera. Although simultaneous recording from both cameras is only available on a few commercial smartphones, this feature does exist and may become more prevalent if desirable use cases are developed. For HOMER, the authors developed applications on the HTC One M8, which is one of the smartphones with the dual camera feature.

Both videos are sent to the web service and fed as inputs to the highlight generation algorithm. Once generated, the highlight(s) are sent back and stored locally on the phone for later browsing in the Android application. A user interface (Figure 15) allows the user to specify highlight duration preferences by setting the parameters $T_{HL}^{m}in$ and $T_{HL}^{m}ax$ mentioned in Section 3.2.

### 4.2. Application 2: Highlights Montage Creation for Library Videos

The second application, also implemented as an Android application, aims to use the highlight generation service to create memorial video albums (Figure 14). Similar to the services proposed by Facebook, Google Photos, or Apple, the idea is to assemble a series of short sequences cut from the user’s original videos into a “Best of” video. The novelty of this implementation is the use of the user’s emotions fused with other features to select video highlights when creating the final montage video.

In order to take advantage of HOMER’s functionalities, the user is asked to choose a time range corresponding to the start and end times of the memory period (Figure 16). The videos from the user’s library matching this time range are loaded into the application ready to be played sequentially so that the user’s face can be simultaneously recorded by the front camera of the phone. After each video is watched, the video and the corresponding facial recording are sent to the web service where the algorithm converts them into highlights. The resulting highlights are shuffled and concatenated into a “Best of” montage video, which is sent back and stored on the user’s mobile device.

Even though applications 1 and 2 harvest the scene video in two different ways (one directly recorded, the other one in a digital library), the videos are handled in the same way, and are sent with the corresponding facial video to the server. When a user rewatches a video, the user is still predisposed to show emotions.

### 4.3. Future Applications

We outline two other applications that can leverage the highlight generation service.

With the emergence of the streaming platform Twitch, watching gamers play video games live is a rapidly growing source of entertainment [65]. Often, streamers include a camera preview of their face in their stream alongside the live recording of the game that they are playing, providing both a facial video and a scene video that can together be processed by the web service to generate highlights. Indeed, a 2016 survey found that Twitch viewers watch streams for approximately 11.0 h per week [66], and currently, Twitch only provides a manual highlight annotation interface. Thus, bringing automatic highlight generation to a streaming platform like Twitch can save significant time on both the streamer’s end when making highlight videos and the viewer’s end when wanting to watch a missed stream. This application would make the frame similarity an irrelevant feature for the system, and would rely on audio and emotional features only.Video calling applications such as FaceTime and Skype canonically involve two facial videos, but one can be treated as the scene video as well. In that case, highlights of a video call can be generated, for example capturing funny jokes or exciting parts of the conversation.

## 5. Evaluation and Discussion

As explained in Section 3.5.1, the ground truth for video highlights is subjective, differs between people, and depends on the presence of a disruptive semantic event. To ensure an objective evaluation of HOMER, we enlisted focus groups composed of people with different backgrounds to help evaluate the system.

### 5.1. Experimental Setup

To validate HOMER’s performance, we conducted an experiment to validate HOMER’s performance against baseline algorithms.

#### 5.1.1. Data Acquisition

For evaluation, we uploaded the automated highlight generation application on an HTC One M8 smartphone, which supports the simultaneous recording of front and back cameras. While this capability is not enabled in some versions of Android and iOS, it is not a limitation of the smartphone hardware, and can be activated by software updates.

In this experiment, a set of 20 videos was randomly chosen from the Video Titles in the Wild (VTW) dataset supported by Microsoft Research [67]. The set constituted personal, non-professional videos taken in everyday contexts, including those that trigger laughter and excitement, as well as those that are more emotionally uniform.

#### 5.1.2. Participants

A set of 20 videos was randomly chosen from the Video Titles in the Wild (VTW) dataset supported by Microsoft Research [67]. The dataset includes non-professional videos taken in everyday contexts, including videos that trigger laughter and excitement, and videos that are emotionally uniform.

Moreover, 10 subjects were recruited for the experiments to promote reliable statistics and human diversity, since emotional responses to a video can vary significantly from person to person. Subjects were aged between 19 and 30, three female and seven male, with varying educational backgrounds (from arts to engineering). Subjects were recruited with IRB approval. In the first experiment, after recording a video from the front and back cameras, subjects were asked to label the video highlights by start and end time. In the second experiment, subjects were asked to watch the set of VTW videos in a fixed randomized order on a mobile device, with their facial reactions being recorded and the video highlights labeled. In addition, subjects were shown the output of HOMER and asked to rate their satisfaction with the generated highlights as “satisfied”, “partially satisfied”, or “not satisfied”.

### 5.2. Results

#### 5.2.1. Metric

The generated highlights were evaluated frame-wise using $F_{1}$ score, which represents the normalized harmonic mean of precision (P) and recall (R):$$F_{1}=\frac{2\;P\;R}{P+R},\;where:\;P=\frac{\mathrm{Nb}\;\mathrm{of}\;\mathrm{frames}\;\mathrm{in}\;A\cap G}{\mathrm{Nb}\;\mathrm{of}\;\mathrm{frames}\;\mathrm{in}\;G},R=\frac{\mathrm{Nb}\;\mathrm{of}\;\mathrm{frames}\;\mathrm{in}\;A\cap G}{\mathrm{Nb}\;\mathrm{of}\;\mathrm{frames}\;\mathrm{in}\;A}$$
where A = Annotated highlights, G = Generated highlights.

#### 5.2.2. Baseline

The Pronounce level model presented by Joho et al. [10], which represents an affective external summarisation technique, was used to establish a baseline for comparison. In their paper, the generated highlight’s duration is a fixed ratio of the overall video length. Across different ratios of 10%, 25%, 50%, and 75%, the Pronounce level model performs similarly to other affective models discussed in Section 2, achieving an $F_{1}$ score within 0.03 of the best model for each ratio. At a high level, the model categorizes emotions into three groups: no, representing neutral emotions; low, representing angry, disgust, fear, and sad emotions; and high, representing happy and surprise emotions. The three groups are given scores of 0, 0.5, and 1 respectively, and the most prominent emotion for a given frame determines the score at that point. A Kaiser window is applied to smooth the score function, and for a fixed highlight ratio, the segment that maximizes the score function is chosen.

In addition to the Pronounce level model, we also implemented variants of HOMER which only utilize frame similarity or audio extraction for highlight extraction. For these two baselines, the implementations are identical to HOMER, except for the inclusion of only a single input source. For example, the frame similarity pipeline calculates the begin/end of highlights as $f_{\alpha }\left(t\right)=f_{FS}^{\alpha }\left(t\right)-\lambda t^{\prime 2}$ and $f_{\beta }\left(t\right)=f_{FS}^{\beta }\left(t\right)$, as in Equations (Equation 1) and (Equation 4).

#### 5.2.3. Highlight Generation

To evaluate highlight generation, 20 videos were taken from the VTW dataset. The video duration ranged from 13.3 to 37.7 s with a mean duration of 21.9 s and a standard deviation of 6.5 s. The 20 videos were classified into three categories—“funny”, “exciting”, and “other”—where the “funny” and “exciting” videos tended to evoke high arousal, while the “other” videos evoked low arousal but high valence. The videos were categorized manually. For example, one “funny” video involved a girl trying to surf in a wave pool but ends up slipping and falling, one “exciting” video captured a man doing a back-flip off a cliff into a pond, and one “other” video showed a baby cuddled up with a dog.

In the evaluation experiment, subjects annotated 26.6% of each video as highlight on average, and our highlight generation algorithm predicted 20.9% of each video as highlight. To bolster the closeness of comparison, the Pronounce level model was tested on the evaluation videos with the fixed ratio of 25%, resulting in an $F_{1}$ score of 0.339. HOMER demonstrates an improvement of 38% with an $F_{1}$ score of 0.469, as shown in Figure 17. As explained in Section 3.5.2, a change of emotional state often occurs in response to a stimulus event, which is a key component in identifying a highlight. By only basing the highlight segmentation on the Pronounce level, it is likely that the Pronounce model will miss the stimulus. In our case, we use audio-visual features to decide how long before the emotional response we should go to optimally capture the stimulus. Audio-visual features also allowed us to detect more precisely the highlight end, which does not necessarily correspond to the end of the emotional response.

Figure 17 displays the mean $F_{1}$ scores for all 20 videos, as well as for each of the three video categories. The generated highlights best matched the labels for the “funny” videos, while they performed worst on the “other” videos. However, this difference in $F_{1}$ score was not completely unexpected. Videos that typically evoke high arousal, e.g. funny videos, tend to retain their potency even when viewed vicariously. In other words, whether a funny video is a personal video or not minimally affects whether someone actually finds it funny. On the other hand, videos that evoke lower arousal but higher valence often require being in the moment to feel the entirety of the emotion. Watching your own dog play with the garden hose is very different from watching a different dog play with a garden hose.

#### 5.2.4. Satisfaction

Finally, we present the cumulative satisfaction ratings for the generated highlights in Figure 18 to provide an alternative angle to evaluate our algorithm. This additional data were collected to account for the flexibility and subjectivity involved in highlight labeling.

Intuitively, the priority is that the generated highlight is satisfactory to the user, as opposed to precisely capturing exactly the highlight that the user has in mind. For example, some users might prefer more context, desiring additional frames to precede the true “highlight event”, but they are satisfied as long as the “highlight event” is captured. Thus, $F_{1}$ score does not provide a complete picture, and supplementing the metric with satisfaction ratings may be beneficial. In the cases where subjects believed that there was at least one highlight, only 25% of generated highlights were not satisfactory, and for the majority of videos, subjects were satisfied with the highlights generated by HOMER.

#### 5.2.5. Discussion

As mentioned in Section 2, there are a number of recent works which have attempted to utilize emotion in addition to video features to generate highlights. Works such as [43,44] produce high quality highlights on football videos and video game streams, but may not perform well on more varied videos, as selected in this work. In addition, [43] requires human input to record emotion, and [44] is designed specifically for streaming setups.

Other works, such as [45,46], require significant amounts of external hardware to gather various physiological signals. Furthermore, their focus is on video ads, which may not be similar to videos captured using a smartphone. Lastly, we compared our work with Joho et al. [10] and showed an improvement in $F_{1}$-score, which can be attributed to incorporating audio and video features, along with emotion.

Machine learning methods for internal highlight generation have been widely explored and have shown good results. In this work, we explored using deep learning to generate start and end timestamps for highlights, as in Section 3.5.1; however, we were able to achieve a better performance using a hand-designed algorithm. One possibility for improving this method is to incorporate additional training data to improve the performance of the classifiers. In future works, machine learning methods will be an important future direction towards achieving even better performance for personalized highlight generation using emotion, audio, and video features.

## 6. Conclusions

We present HOMER, a cloud-based system for video highlight generation. Our hybrid approach achieves high performances and outperforms state-of-the-art solutions by fusing internal video content-based features with user’s emotion data. Our perception-based solution makes it possible to detect user relevant video highlights. In two separate experiments, including videos filmed with a dual camera setup, and home videos randomly picked from Microsoft’s VTW dataset, HOMER demonstrates an improvement of up to 38% in $F_{1}$-score from baseline, while not requiring any external hardware. Both the design of the events timeline, synchronizing multi-modal features, and the highlight detection algorithm makes our model expandable and enables the possibility of adding new features. Our platform, available as a web service, provides an API with parameters for highlight generation. These settings allow the user to either specify highlight duration constraints or let HOMER automatically find an optimal duration. Another parameter enables the capture of multiple highlights from a single video, increasing the adaptability of the system and the range of potential applications. The ubiquity of smartphones with front-facing cameras makes our system accessible for future applications, as we demonstrated through the implementation of two smartphone applications. HOMER addresses the challenge of video overload by intelligently filtering and creating shorter highlights that are enjoyable for the user.

## Figures and Tables

**Figure 1 sensors-21-01035-f001:**
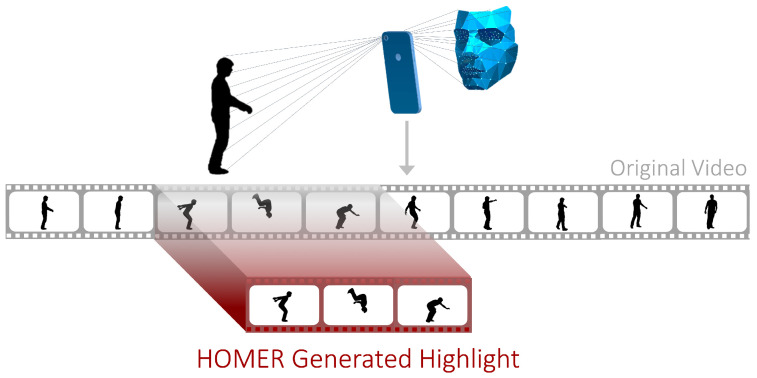
HOMER: a multi-modal video highlight generator based on emotion recognition.

**Figure 2 sensors-21-01035-f002:**
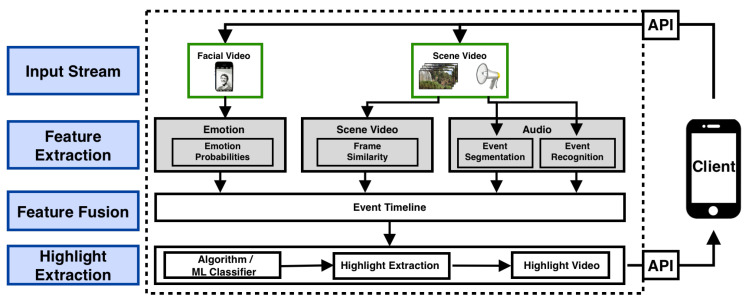
HOMER System Architecture.

**Figure 3 sensors-21-01035-f003:**
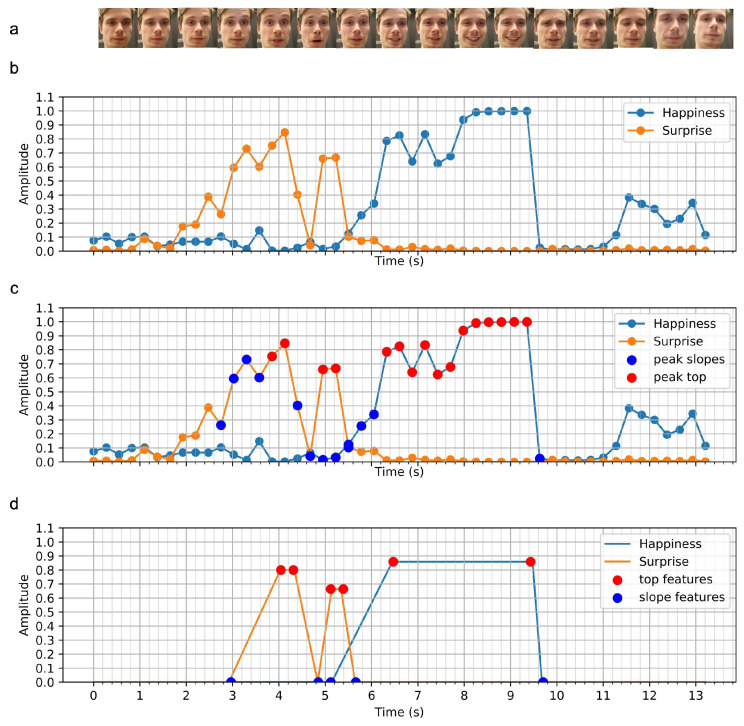
Raw signals to features pipeline: (**a**) frames after face detection, (**b**) prediction probabilities of happiness and surprise made by the neural network model (**c**) peak detection and characterization performed on the predictions (**d**) extracted peak features.

**Figure 4 sensors-21-01035-f004:**
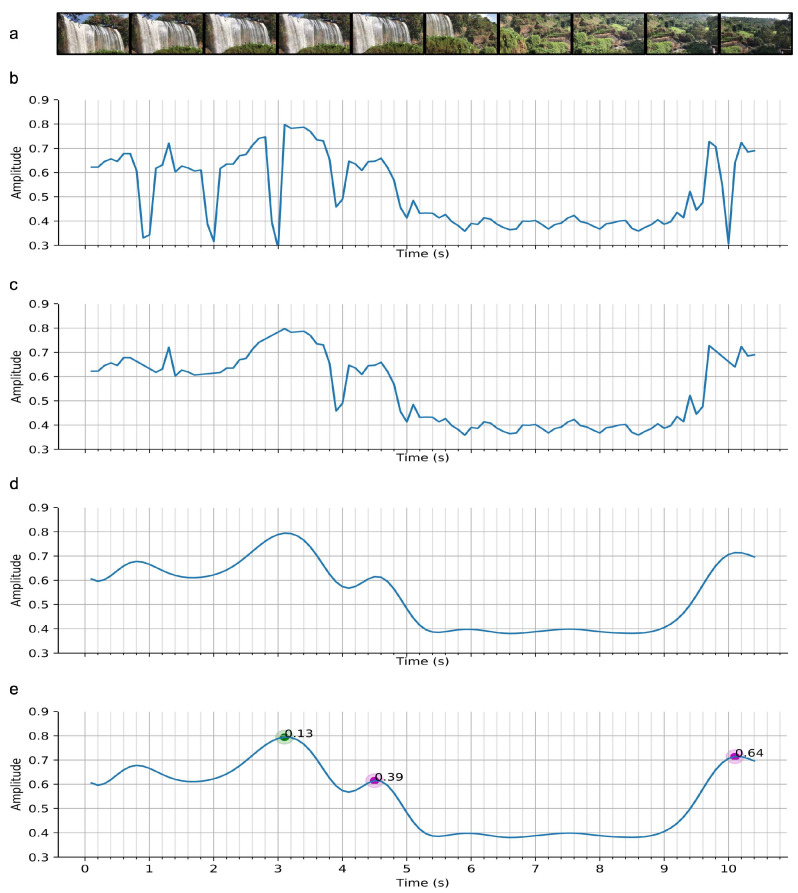
Raw signal to features pipeline: (**a**) Frames extracted from the scene video, (**b**) Frame similarity from Zhang et al. [54], (**c**) discontinuity removal, (**d**) 2.5Hz low-pass filter, (**e**) rule-based feature extraction.

**Figure 5 sensors-21-01035-f005:**
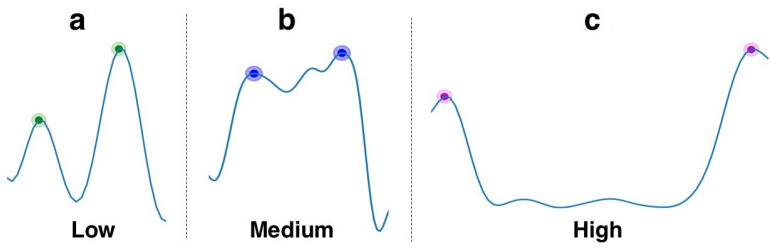
Three main frame similarity (FS) patterns encountered and their importance: (**a**) Single peaks, (**b**) High similarity plateau ≡ static camera, (**c**) Low similarity valley ≡ camera pan.

**Figure 6 sensors-21-01035-f006:**
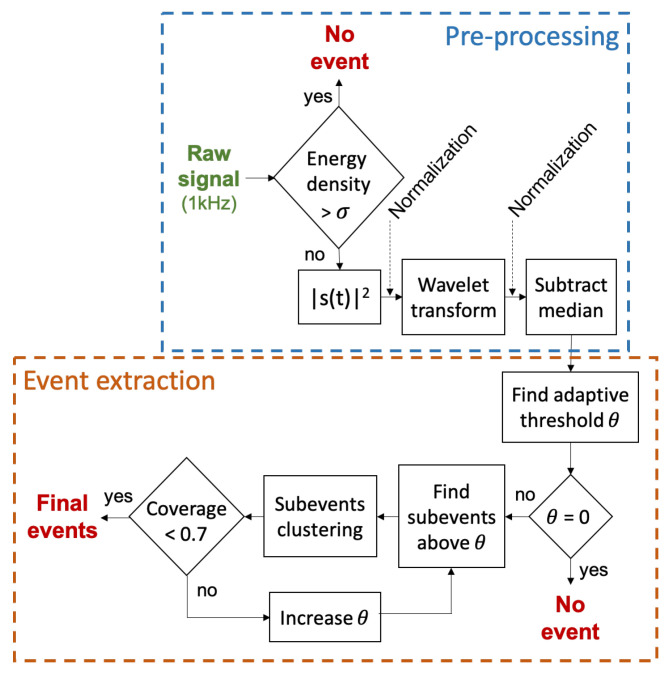
Diagram of the overall sound events detection algorithm.

**Figure 7 sensors-21-01035-f007:**
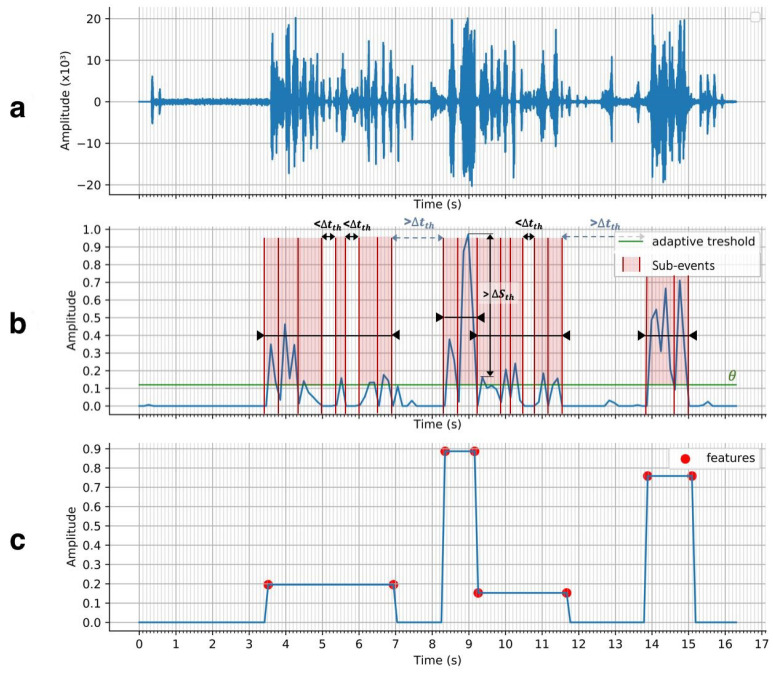
Representation of the sound signal at three main stages during the sound event detection process: (**a**) raw sound signal, (**b**) pre-processed signal, (**c**) sound event features after event extraction.

**Figure 8 sensors-21-01035-f008:**
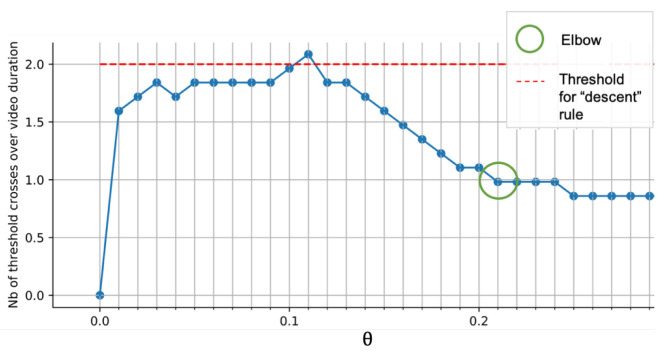
Adaptive threshold search: number of crosses of the threshold Θ by the sound signal.

**Figure 9 sensors-21-01035-f009:**
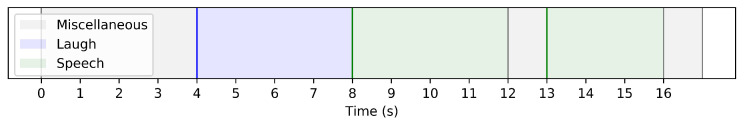
Sound event classification performed on the whole sound sample with three classes: speech, laughter and miscellaneous.

**Figure 10 sensors-21-01035-f010:**
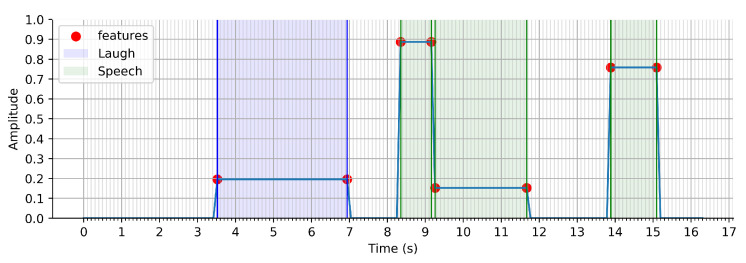
Fusion between event segmentation and event recognition.

**Figure 11 sensors-21-01035-f011:**
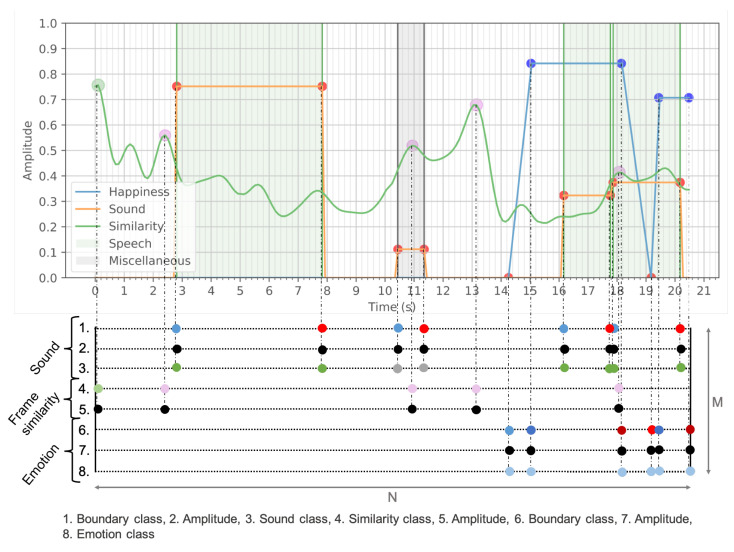
Representation of the events timeline built upon the extracted features.

**Figure 12 sensors-21-01035-f012:**
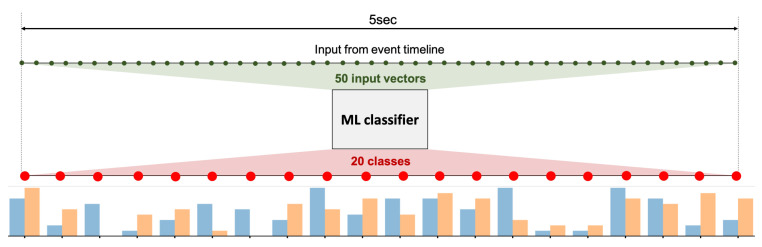
Diagram illustrating the input and output of the machine learning (ML) classifier: highlight start score function (**orange**) and highlight end score function (**blue**).

**Figure 13 sensors-21-01035-f013:**
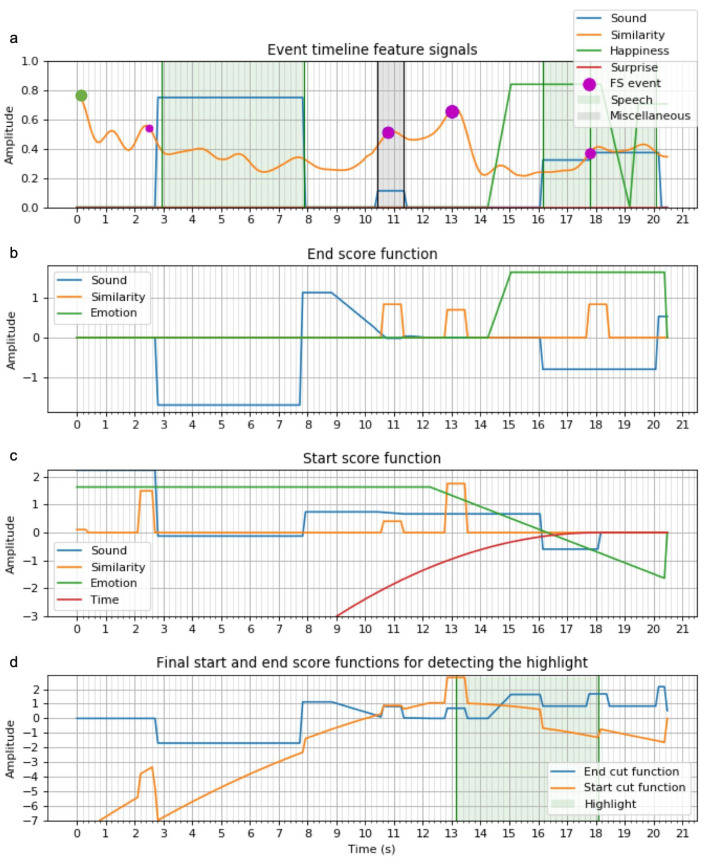
Score functions computation processes to derive both start and end highlight cuts: (**a**) events timeline signals, (**b**) sub-functions to compute end cut score function, (**c**) sub-functions to compute start cut score function, (**d**) resulting start and end score functions for highlight classification

**Figure 14 sensors-21-01035-f014:**
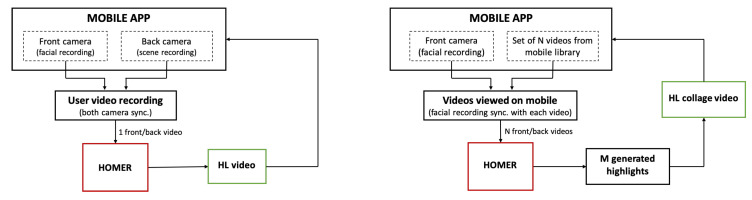
Block-diagrams of Application 1 (**left**) and Application 2 (**right**).

**Figure 15 sensors-21-01035-f015:**
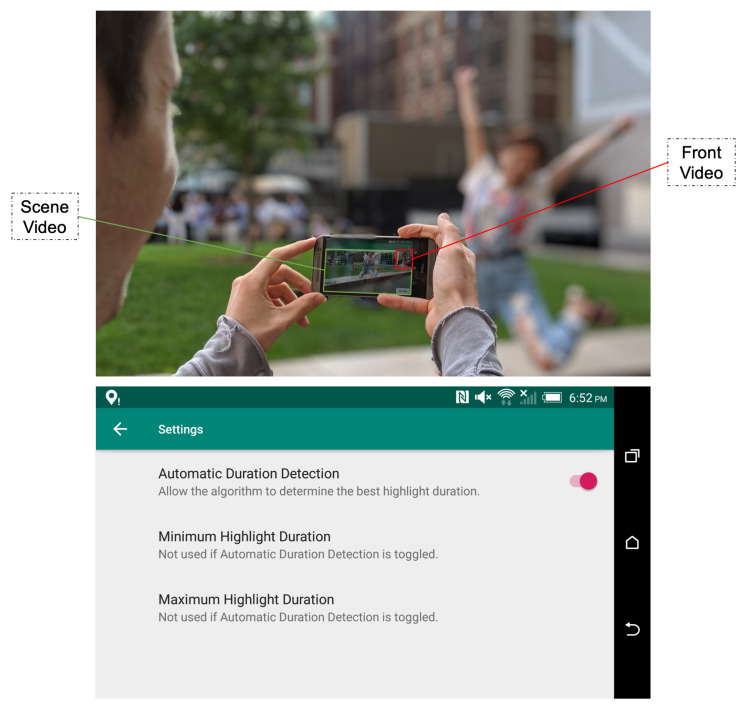
Application 1: photo of the application in use (**up**) and screenshot of the application settings menu (**down**).

**Figure 16 sensors-21-01035-f016:**
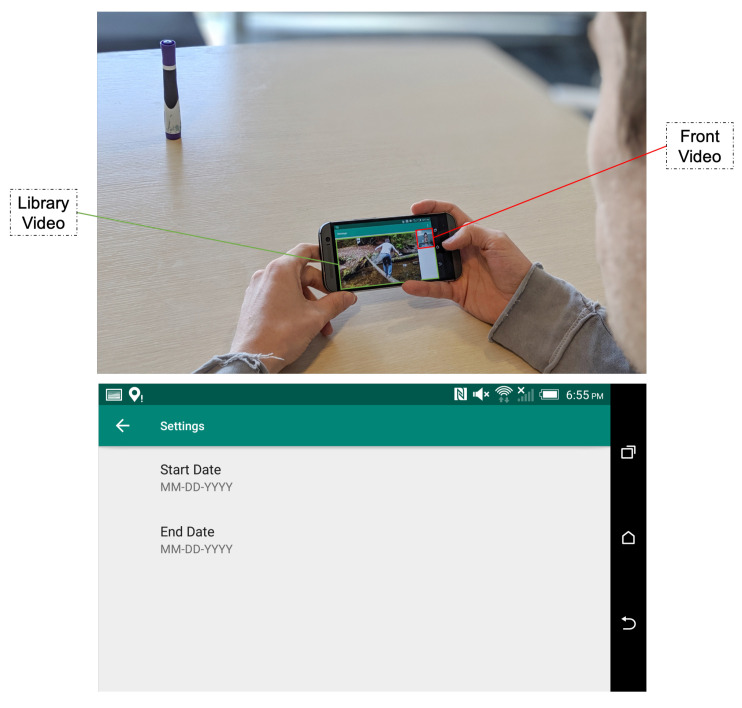
Application 2: photo of the application in use (**up**) and screenshot of the application settings menu (**down**).

**Figure 17 sensors-21-01035-f017:**
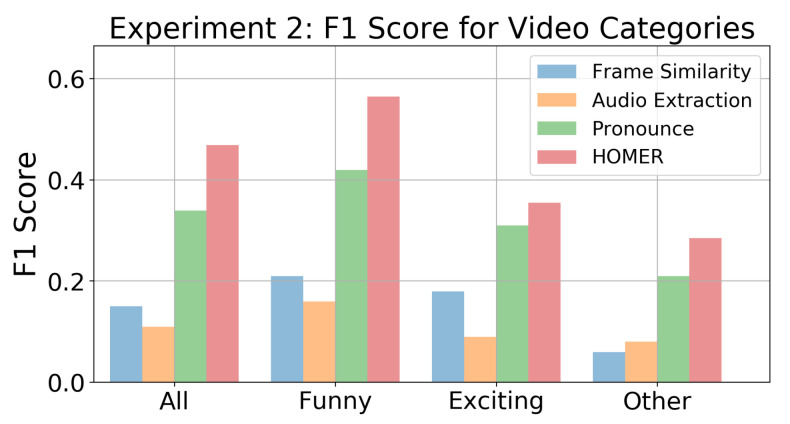
The mean $F_{1}$ scores of 20 VTW videos for different video categories.

**Figure 18 sensors-21-01035-f018:**
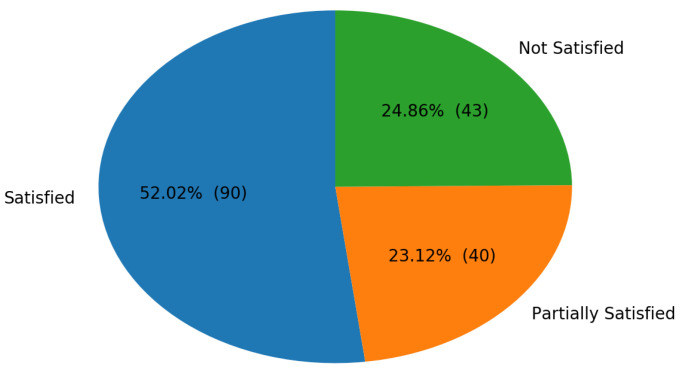
Proportions of the three highlight generation performance categories: satisfied, partially satisfied, and not satisfied.

**Table 1 sensors-21-01035-t001:** Keys of limitations for Table 2.

Key	Limitation
1	Videos Scope
2	Audio/Visual Features Only
3	External Hardware
4	Not Mobile Platform

**Table 2 sensors-21-01035-t002:** Limitations of existing studies in highlight generation.

Source	Year	Sensing Inputs	Summarization Category	Limitations
Fiao et al. [43]	2016	Emotions, Audio, Video	Hybrid	1 (sports), 3, 4
Yang et al. [17]	2015	Audio, Video	Internal	2, 4
Shukla et al. [27]	2018	Audio, Video	Internal	1 (sports), 2, 4
Kaklauskas et al. [45,46]	2018, 2019	Audio, Video, Eye tracking, Facial Video, IR Camera, Personalized Questionnaire	Hybrid	1 (video ads), 3, 4
Gunawardena et al. [47]	2020	Video	Internal	2, 4
Zhang et al. [48]	2020	Video	Internal	2, 4
Moses and Balachandran [49]	2018	Video	Internal	1 (surveillance), 2, 4
Ringer and Nicolaou [44]	2018	Emotions, Audio, Video	Hybrid	1 (video games), 4
Joho et al. [10]	2009	Emotion	External	3, 4
Chênes et al. [42]	2012	Skin Temperature EMG, EDA, BVP	External	1 (Movies), 3, 4

**Table 3 sensors-21-01035-t003:** Results of highlight start/classification for different models using accuracy.

	Rand.	LR	SVM	kNN	RF	Adaboost
Acc.	0.05	0.13	0.10	0.10	0.12	0.12
Acc. (±0.5 s)	0.20	0.35	0.27	0.35	0.31	0.32

## Data Availability

Code developed in this work can be found at https://github.com/Columbia-ICSL/HOMER.git.

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
