# Peer review of "Intelligent Video Highlights Generation with Front-Camera Emotion Sensing"

_sensors, 2021, doi:10.3390/s21041035_

Round 1

Reviewer 1 Report

This manuscript contains exciting work from the intelligent video highlights generation with front-camera emotion sensing. However, the manuscript is written in style more like a report rather than a research article. The global innovativeness in research development hasn't been presented. Some figures and tables which involve world-wide novel research should be described and discussed with more details to emphasize the state-of-the-art-review all over the world novelty. Please use this the newest (2017-2020) Web of Science journals papers, such as:

Kaklauskas, A., Bucinskas, V., Vinogradova, I., Binkyte-Veliene, A., Ubarte, I., Skirmantas, D., & Petric, L. (2019). INVAR Neuromarketing Method and System. Studies in Informatics and Control, 28(3), 357-370.

Kaklauskas, A., Zavadskas, E. K., Banaitis, A., Meidute-Kavaliauskiene, I., Liberman, A., Dzitac, S., ... & Naumcik, A. (2018). A neuro-advertising property video recommendation system. Technological Forecasting and Social Change, 131, 78-93.

Reviewer 2 Report

This paper presents a cloud-based system for video highlight generation. The paper is well written, the research appears to be sound, the results are exciting for their potential that can leverage the highlight generation service, the study will be of interest to Sensors readers, and I believe that the paper merits publication in Sensors.

However, I do have some comments given as follows:

- It would be good to make a comparison between the results obtained by the authors and other similar solutions with hybrid approaches or not.
- The way the figures are placed in the text should be standardized, either before or after the reference in the text (preferably after).
- In Figure 2, all the three blocks from the Highlight extraction level should be indicated inside of a rectangle, so the arrow will point to all functional blocks (not only to Highlight extraction block).

Reviewer 3 Report

The subject is interesting but the following aspects must be clarified:

  • explain more clearly why machine learning are unsuitable (eg. unsupervised methods)
  • Section 5.2.3 - how were classified the videos (manually or automatic - in case of automatic what method was used)
  • comparison with other existing methods will be useful
  • Figure 1 must be placed after its appearance in the text (the same for figure 3, 6, 7, ...). The same for tables.
  • Section 3 starts with a figure. It must start with a paragraph, describing the figure 2. 

Round 2

Reviewer 1 Report

I want to recommend using state-of-the-art sensors for future research.

Reviewer 3 Report

All my comments were addressed and I recommend to publish the paper.